# MASKED AUTOENCODER FOR GRAPH CLUSTERING WITHOUT PRE-DEFINED CLUSTER NUMBER K.

## ABSTRACT

Graph clustering algorithms with autoencoder structures have recently gained popularity due to their efficient performance and low training cost. However, for existing graph autoencoder clustering algorithms based on GCN or GAT, not only do they lack good generalization ability, but also the number of clusters clustered by such autoencoder models is difficult to determine automatically. To solve this problem, we propose a new framework called *G*raph *C*lustering with *M*asked *A*utoencoders (*GCMA*). It employs our designed fusion autoencoder based on the graph masking method for the fusion coding of graph. It introduces our improved density-based clustering algorithm as a second decoder while decoding with multi-target reconstruction. By decoding the mask embedding, our model can capture more generalized and comprehensive knowledge. The number of clusters and clustering results can be output end-to-end while improving the generalization ability. As a nonparametric class method, extensive experiments demonstrate the superiority of *GCMA* over state-of-the-art baselines.

## 1 INTRODUCTION

Graph clustering is an important unsupervised learning task. Moreover, in the unsupervised (more realistic) setting, the number of classes (denoted by $k$) and their relative sizes (i.e., class weights) are unknown. With the development of deep learning techniques, many deep learning based methods are able to cluster large rows of high-dimensional datasets more efficiently, still not bypassing this problem. Most of the existing high-performance methods are parametric class methods, either for the classical clustering problem for large image datasets (Xie et al., 2016), or for clustering more complex graph-structured datasets (Kipf & Welling, 2016; Hasan-zadeh et al., 2019). Non-parametric methods are few and far between.

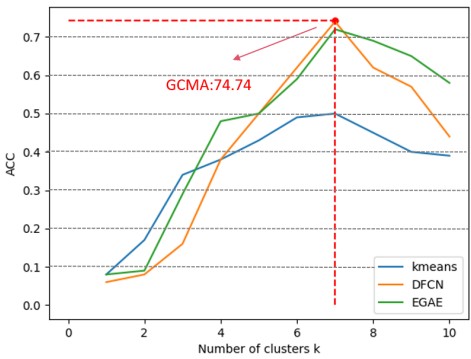

Figure 1: The selected algorithm on the Cora dataset includes the current SOTA method based on the trend of ACC changes in the number of clusters $k$.

However, it is clear that non-parametric class methods (Ronen et al., 2022) are more advantageous in real-world use. This is because the automatic determination of the correct $k$-value by non-parametric methods can have a positive effect on the clustering performance, e.g., through the process of dividing individual clusters into multiple clusters whose clustering labels change accordingly. This may lead to convergence to better local optima and performance gains (Chang & III, 2013). Also, good clustering results are based on determining the correct $k$-value, if the $k$-value is wrong even with the sota method, the performance becomes poor as in Fig. 1. It is clear to see that even with the current SOTA method, without the correct value of $k$ its performance plummets.

Also for graph data, the high dimensionality and complex graph topology further increase the difficulty of clustering. Most of the existing methods (Zhang et al., 2019; Zhang & Li, 2023; Wang et al., 2019a) attempt to cluster after learning the representation first by generating perspectives,

such as clustering based on graph autoencoder structure (GAE), which treats the graph input itself as self-supervised and learns to reconstruct the graph (Wang et al., 2019a). However, literatures (Hassani & Ahmadi, 2020; Velickovic et al., 2019; Zhu et al., 2020) suggest that GAE following this simple graph reconstruction principle may overemphasize proximity information, which is not always beneficial for self-supervised learning. Therefore, designing better excuse tasks is the key to improving the quality of learned graph embedding.

Masked autocoding has been successfully applied in both NLP and CV domains, with prominent examples being BERT (Devlin et al., 2019) and MAE (He et al., 2022), respectively. In fact, masked autocoding should also be well suited for graph data, since both edges and nodes can be easily masked or unmasked as self-supervised, as demonstrated by existing studies (Shi et al., 2020; Ma et al., 2018). Whereas its strong generalization ability is more conducive to learning good graph embeddings, few relevant studies have confirmed its performance in clustering.

In our work, we use a autoencoder that learns graph embeddings mainly in a graph masking manner to reconstruct the structural decoding after the graph embeddings, so that the learned representations have better interpretability. The generalization capability is then increased by introducing our improved density-based clustering algorithm as a second decoder, which automatically determines the appropriate $k$-value. Our main contributions are as follows:

- To the best of our knowledge, this is the first work to apply a graph masking autocoder to a clustering task, and the first method to determine the number of clusters specifically for graph data.
- Our model uses the mask graph mechanism to have better generalization ability and interpretability. This allows learned representations to be applied to multiple types of downstream tasks
- Extensive experiments on five datasets demonstrate that our model outperforms existing state-of-the-art baselines.

## 2 RELATED WORK

### 2.1 NONPARAMETRIC DEEP LEARNING CLUSTERING METHODS

Among the few non-parametric classes of methods(Shah & Koltun, 2018; Zhao et al., 2019; Wang et al., 2022; Ronen et al., 2022; Wang et al., 2018) for predicting the number of clusters, we consider the most representative to be DeepDPM (Ronen et al., 2022) and DED (Wang et al., 2018). DED utilizes the advantages of feature representation learning techniques and density-based clustering algorithms. It uses a deep convolutional autoencoder (Walker et al., 1984) and t-SNE to learn appropriate embedding features, and then predicts the number of clusters using a new density-based clustering algorithm (Rodriguez & Laio, 2014). However, DED belongs to the classical two-step algorithm, which does not output end-to-end results with the value of $k$. DeepDPM is based on the Dirichlet process and adapts to the dynamic architecture of judging the changing $k$ using a split/merge framework. It verified performance on the ImageNet dataset and is the current SOTA method. However, this method is not applicable to graph data due to its inability to capture graph topology information. And it also has some limitations in terms of interpretability and generalization ability.

### 2.2 MASKED AUTOENCODERS FOR GRAPH

The learning task of masking autocoding is to mask the part of the input information and predict the hidden content. This is a self-supervised learning method. Masked language modeling (MLM) (Devlin et al., 2019) is the first successful application of masked autocoding in natural language processing. Its working principle is similar to the cloze test in English. Recently, masked image modeling (MIM) (He et al., 2022) follows a similar principle by masking redundant pixel blocks and predicting them for learning. Although masked autocoding technology is very popular in language and visual research, it is relatively less studied in the graph domain. In the work of the first graph masking automatic encoder, MGAE (Tan et al., 2023) tried to implement the masking strategy on the edges of the graph structure as a self-supervised learning paradigm. MaskMAE (Li et al., 2023)

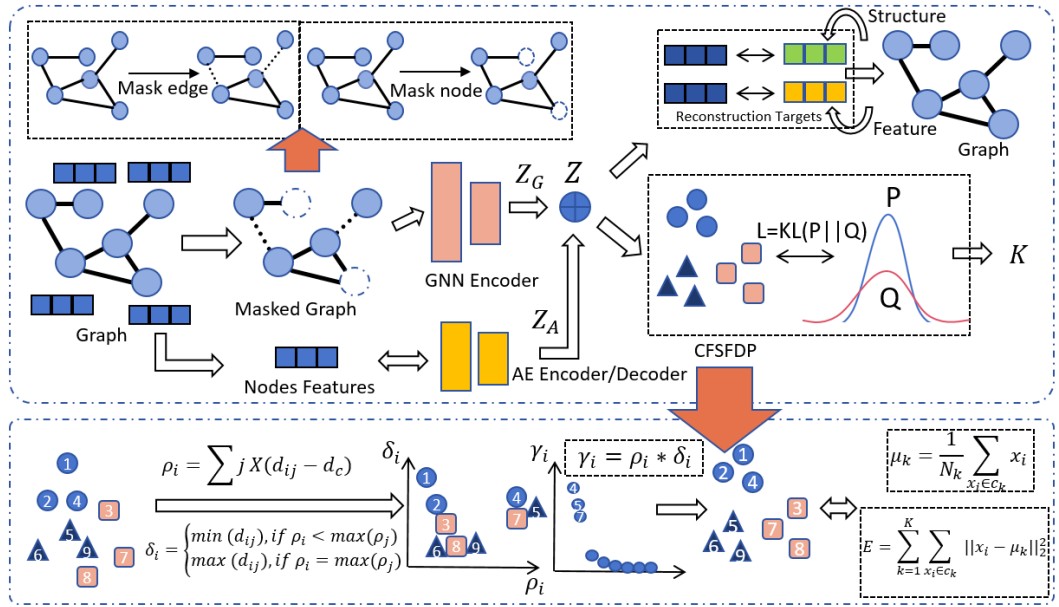

Figure 2: Flowchart of GCMA. The top half represents the overall model architecture and the bottom half represents our improved clustering algorithm process.

adds a new one-sided mask and path mask strategy and also adds a theoretical analysis of the side mask strategy. Similar to us, the GiGaMAE (Shi et al., 2023) method masks both nodes and edges. However, they focus more on how to improve generalization ability and lack some improvements for interpretability and clustering tasks.

### 2.3 DEEP GRAPH CLUSTERING

For graph data, the long-standing research direction lies in single-structure graph autoencoders, mainly GNNs. They utilize GCN or GAT structures and apply convolutional structures or attention mechanisms to graph structures. Node weights are assigned through graph topology specific information, which in turn leads to graph embedding representations. For example, (Kipf & Welling, 2016; Wang et al., 2019a; Zhang & Li, 2023). However, some scholars believe that such a structure will cause the model to pay too much attention to the graph topology information and lose the node feature information, so they proposed a fusion graph autoencoder structure. Their basic idea is to fuse the linear neural network layer with the coding layer of the graph autoencoder to get a more powerful fusion representation. The difference lies in the fusion mechanism. For example, the mechanism used in Bo et al. (2020) is a linear fusion between layers, while Tu et al. (2021) uses a deep fusion mechanism under triple validation. Compared to our improved masked graph autoencoder, as revealed in the literature (Hassani & Ahmadi, 2020; Velickovic et al., 2019), such GAE structure class methods lead to a disproportionate ratio of learned structural, proximity and feature information, resulting in limiting the learning of graph embeddings to a large extent. What's more, they all belong to parameter class methods, which need to set the correct $k$ value in advance.

## 3 METHOD

Our framework is shown in Fig. 2 and consists of 3 parts: the mask fusion network consists of an encoder, a decoder with multiple reconstruction targets and a modified density-based algorithm as a second decoder, with the graph embedded in a self-optimizing module.

### 3.1 Masking Fusion Encoder

#### 3.1.1 Graph Mask Encoder

Our graph mask encoder consists of 2 components: graph mask and encoder. The first thing that needs to be explained is the graph mask and encoder. The original graph $G = \{V, A, X\}$ is used as the input and is masked into $G' = mask(G)$ by edge masking (Tan et al., 2023) or feature masking (Wang et al., 2017). $V$ denotes the set of nodes, $A$ denotes the adjacency matrix and $X$ denotes the node features. The encoder can adopt GAT (Velickovic et al., 2017) and GCN (Kipf & Welling, 2017) architectures. The encoder takes $G'$ as the input, encodes the nodes into a representation graph, and embeds it to obtain a low-dimensional feature representation. Finally, the decoder is reconstructed from the graph embedding.

It is worth noting that masking both modal information at the same time will have a negative impact on model learning, because it may lack enough information for graph reconstruction. However, only masking and reconstructing one mode limits the ability of the model to learn from another mode, which hinders the learning of comprehensive representation. GiGaMAE (Shi et al., 2023) has found a multi-objective reconstruction that can mask both edge and node features of the original graph during the training process. Formally, we denote the mask process by $M$, $M = \{M_e, M_f\}$. Where the edge mask matrix $M_e \in \{0,1\}^{|V| \times |V|}$ and the feature mask matrix $M_f \in \{0,1\}^{|V|}$ are randomly generated binary matrices. The perturbation can be controlled by the sparsity of $M_e$ and $M_f$. In this way we can obtain $M(G = \{V, A, X\}) = \{V, A * M_e, X diag(M_f)\} = \{V, A', X'\} = G'$, where $A'$, $X'$ and $*$ respectively represent the mask feature matrix, the mask adjacency matrix and the Hadamard product. We refer to these nodes, which are masked with edge or node characteristics, as $\widetilde{V}$. Enter the mask map into the encoder to obtain the potential representation $Z_m$.

#### 3.1.2 Feature Encoding and Integration mechanisms

Considering the problems mentioned in Chapter 1, the graph autoencoder structure will overemphasize the proximity information and distort the node feature information. Therefore, here we introduce a simple AE network dedicated to extracting the feature information of the nodes and can also be used as a reconstruction target to fully utilize the node features of the graph. Where each layer of AE calculates the node features are represented as follows, $w^l$ and $b^l$ are hyperparameters:

$$Z_{ae}^l = f(w_l z^{l-1} + b^l), \tag{1}$$

We use a linear fusion mechanism to fuse the two graph embeddings to obtain a more complete and robust representation. $\epsilon$ is the learnable coefficient which is automatically adjusted according to the gradient fitting method, thus adjusting the importance of the two embedded parts. In this work, $\epsilon$ is initialized to 0.1:

$$Z = (1 - \epsilon)Z_m + \epsilon Z_{ae} \tag{2}$$

Technically, our cross-modal linear fusion mechanism takes into account sample correlation in terms of node characteristics and proximity information. Therefore, it has potential advantages in finely fusing and refining cross-modal information to learn potential representations.

### 3.2 Multi Target Decoder

#### 3.2.1 Masking Graph decoding

Different graph models focus on different focuses of graph information. For example, GAE prioritizes learning graph structure information, while AE trained on feature matrix X mainly encodes graph node attribute information. Therefore we consider embedding reconstruction methods that aim at fusing multiple models such as graph models. The loss function is constructed based on mutual information (MI). Here $Z_n \in R^{|V| \times d_n}$ is used to denote the $n - th$ reconstruction objective. The information from various modalities is stored in the chi-square continuous embedding space $Z_n$.

We remask (Hou et al., 2022) $Z_m$ in order to obtain more node compression denoted $\hat{V} \in R^{|V| \times d}$. This way we can represent each node $v_i$ as $\hat{v}_i = \hat{V}[i, :] = encoder(A', X')$. At the same time, We

remask $Z_n$ to get $S_n$. Let $S_n[i,:] = s_i^n$, then $s_i^n = Z_n[i,:], v_i \in \widetilde{V}$. We implement the decoder with a set of mapping functions $P$, which are designed to map $\hat{V}$ to the target embedding space $S_n$. Each mapping is implemented as a multilayer perceptron (MLP). So we have $\hat{V}_n = P(\hat{V}) \in R^{|v| \times d}$.

When reconstructing the targets, $l2 - norm$ loss considers the characteristics of each target alone ignoring the relationship between them. While cross entropy is more suitable for discrete variables. Since we are computing the reconstruction loss with multiple targets, we need to capture as much useful information as possible from the targets. So we choose to use MI as the basis for calculating the loss. Maximize the compression representation $\hat{v}$ with target embedding $S_n$. For two targets the reconstruction loss function $L_p$ is:

$$L_p = \lambda_1 P_1(\hat{V}, S_1) + \lambda_2 P_2(\hat{V}, S_2) + \lambda_3 P_3(\hat{V}, \{S_1, S_2\}) \tag{3}$$

$\lambda$ is the parameter that regulates the importance of the information source. And we set $\lambda \geq 0$. The specific calculation steps in Eq.(3) are as follows. $\xi$ is the temperature hyperparameter. $P_n$ represents the mapping for different $S_n$. $\oplus$ denotes concatenation.

$$P_3(\hat{v}, (s_1, s_2)) = exp(\frac{1}{\xi} \times \frac{P(\hat{v}) \times [s_1 \oplus s_2]}{||P(\hat{v})|| \times ||[s_1 \oplus s_2]||}), P_1(\hat{v}, s) = exp(\frac{1}{\xi} \times \frac{P(\hat{v}) \times s}{||P(\hat{v})|| \times ||s||}) \tag{4}$$

For the choice of reconstruction target, we choose to reconstruct the GAE model with the AE model. First we use the output of GAE as our structural target embedding. Given graph G as input, GAE preserves topological information and target embedding by reconstructing the graph structure. Then we use the reconstructed feature structure of the AE part as the reconstruction target to learn the feature information of the input graph. Similarly we can learn the graph node feature information. It should be noted that we linearly fuse the graph embedding of the AE layer in the encoding process, so we also need to take the reconstruction loss $L_a = \sum_{i=1}^{n} ||X - X_a'||$ of the AE part into account. $X_a'$ represents the reconstructed node characterization information.

### 3.2.2 CFSFDP DECODER

The peak density-based clustering algorithm (CFSFDP) (Rodriguez & Laio, 2014) has two main criteria: local density ($\rho$) and minimum distance ($\delta$) to higher densities. One is that data points near the clustering center have lower $\rho$, and the other is that data points are farther away from other denser $\delta$. However, the original algorithm still requires observation to determine the threshold value through experience. Whereas for clustering centers, $\rho$ values are usually much higher than their thresholds, and $\delta$ values may be very close to the thresholds in some cases. Therefore, we judge the range of values of the threshold by many iterations, while using the Gaussian function instead of the indicator function to make the $\delta$ value as much higher than its threshold as possible. Thus the clustering center can be clearly separated from the rest of the data points to get the best $k$ value.

We need smaller spacing within the same cluster as a goal to guide the network to influence the distribution of learning data. Therefore, the mean square error of each sample vector to the corresponding cluster mean vector is calculated, and the mean vector of sample points within each cluster is $\mu_k = 1/N_k \sum_{x_i \in C_k} x_i$, where $C_k$ denotes cluster $k$ and $N_k$ is the number of samples contained in cluster $C_k$. Thus the loss function is $L_c = \sum_{k=1}^{K} \sum_{x_i \in C_k} ||x_i - \mu_k||_2^2$.

### 3.3 SELF-OPTIMIZING MODULES

We used a self-supervised method to optimize the fused graph embedding. We use the t-distribution of Students (van der Maaten & offrey Hinton., 2008) to measure it. The similarity between the node embedding $Z$ and the clustering center embedding was first measured using the $q_{iu}$ metric.

$$q_{ij} = \frac{(1 + ||z_i - \mu_j||^2/v)^{-1}}{\sum_K (1 + ||z_i - \mu_k||^2/v)^{-1}} \tag{5}$$

Thus our optimized affiliation matrix $p$ is constructed based on the obtained $q$ as follows.

$$p_{ij} = \frac{q_{ij}^2/\sum_{i=1}^{n} q_{ij}}{\sum_{j'} q_{ij'}^2/\sum_{i=1}^{n} q_{ik}} \tag{6}$$

To minimize the clustering loss, we construct the loss using the KL scatter:

$$L_s = KL(P||Q) = \sum_i \sum_j p_{iu} log \frac{p_{iu}}{q_{iu}} \tag{7}$$

The above method is considered to be a self-supervision mechanism that forces the current distribution $Q$ to converge towards the target distribution $P$ by minimizing the KL divergence loss between the $Q$ and $P$ distributions. We consider fusion coding as a Q-distribution. This can make the spacing between points within the same cluster smaller and the spacing between clusters larger.

### 3.4 JOINT EMBEDDING

Our model is end-to-end, so the above modules can jointly optimize graph embedding and clustering learning. The total objective function is defined as:

$$L = L_p + \alpha L_a + \beta L_c + \gamma L_s \tag{8}$$

Where $\alpha, \gamma$ and $\beta$ are used to balance the various losses. The overall algorithm is shown in Algorithm 1.

---

**Algorithm 1** GRAPH CLUSTERING WITH MASKED AUTOENCODERS (GCMA)

---

**Input:** Graph $G$ with $n$ nodes; Number of iterations $Iter$; Number of layers $Lay$.
    **while** max iterations$< iter$ or convergence **do**
        Randomly masking $G$ as $G'$;
        Encoding of data according to the encoder. This includes performing fusion coding;
        Decoding of multiple reconstruction targets according to Eq. (3);
        Overall optimization after self-optimization process and CFSFDP process according to Eq. (8);
    **Return:** Clustering results,cluster number $k$ and hidden embedding $Z$.

---

## 4 EXPERIMENTS

### 4.1 DATASET AND BASELINE

In our experiments, we evaluated the proposed algorithm on four popular public datasets, including three graph datasets (cora, citeseer (Sen et al., 2008), and DBLP (Wang et al., 2019b)) and a larger dataset (ogbn-arxiv (Hu et al., 2020)), as shown in Table 1.

In our experiments, we compared a variety of algorithms with our method. First, there are four nonparametric methods. They are MTL (Pimentel & de Carvalho, 2020), DNB (Wang et al., 2022), DED (Wang et al., 2018), and DeepDMP (Ronen et al., 2022). It should be noted that these methods are not specifically de-

Table 1: Information of dataset

| Dataset | Nodes | Features | Clusters |
|---|---|---|---|
| Cora | 2708 | 1433 | 7 |
| Cite | 3312 | 3703 | 6 |
| DBLP | 4058 | 334 | 4 |
| Ogbn-arxiv | 169343 | 128 | 40 |

signed for graph data clustering, so we need to make some necessary changes during the runtime. In addition to this, there are other parametric graph clustering algorithms that we mainly use to compare our clustering performance. They are: DNGR (Cao et al., 2016), TADW (Yang et al., 2015), ARGE (Pan et al., 2018), ARVGE (Pan et al., 2018), AGC (Zhang et al., 2019),DAEGC (Wang et al., 2019a),EGAE (Zhang & Li, 2023), SDCN (Bo et al., 2020), DFCN (Tu et al., 2021). In this paper, three metrics, clustering accuracy (ACC), normalized mutual information (NMI) and adjusted rand index (ARI), are used to validate the performance of various models.

### 4.2 COMPARISON WITH STATE-OF-THE-ART METHODS

We first compared the accuracy of predicting $k$ values with non-parametric methods. As shown in Table 2. We ran it 10 times on the dataset and came up with the accuracy and average for $k$ value prediction. It can be seen that our method performs much better than other methods.

Table 2: This is a comparison of the experimental results with the baseline method. We compute the accuracy as the number of times we get the correct result after 10 runs on the dataset. The mean value is the average of the predicted $k$ values.

| Methods | MtL | | DNB | | DED | | DeepDPM | | GCMA | |
|---|---|---|---|---|---|---|---|---|---|---|
| | mean | Accuracy | mean | Accuracy | mean | Accuracy | mean | Accuracy | mean | Accuracy |
| Cora | 3.0(±1.0) | 0/10 | 6.0(±0.0) | 0/10 | 10.2(±0.7) | 0/10 | 5.1(±2.3) | 1/10 | 7.0(±0.3) | **10/10** |
| Citeseer | 6.7(±3.0) | 1/10 | 5.0(±0.0) | 0/10 | 8.0(±2.0) | 1/10 | 4.0(±5.2) | 2/10 | 5.9(±0.6) | **10/10** |
| DBLP | 2.1(±3.8) | 0/10 | 2.0(±0) | 0/10 | 2.8(±2.1) | 0/10 | 3.0(±11.3) | 0/10 | 3.7(±0.8) | **8/10** |
| Ogbn-arxiv | 20.4(±12.1) | 0/10 | 20.0(±0) | 0/10 | 32.0(±14.9) | 0/10 | 50.0(±20.3) | 0/10 | 48.2(±10.1) | **3/10** |

Table 3: This is a performance comparison with the best graph clustering algorithms(%).

| dataset | m1 | DNGR | TADW | ARGE | ARVGE | AGC | DAEGC | EGAE | SDCN | DFCN | GCMA-A | GCMA |
|---|---|---|---|---|---|---|---|---|---|---|---|---|
| | ACC | 41.91 | 56.03 | 64.00 | 63.80 | 68.92 | 70.40 | 72.42 | 71.00 | 74.02 | 73.82 | **74.74** |
| Cora | NMI | 31.84 | 44.11 | 44.90 | 45.00 | 53.68 | 52.80 | 53.96 | 50.25 | 53.90 | 58.00 | **59.16** |
| | ARI | 14.22 | 33.20 | 35.20 | 37.40 | - | 49.60 | 47.22 | 47.02 | 48.10 | 53.04 | **55.41** |
| | ACC | 32.59 | 45.48 | 57.30 | 54.40 | 67.00 | 67.20 | 67.42 | 66.00 | **69.50** | 67.20 | 67.30 |
| Citeseer | NMI | 18.02 | 29.14 | 35.00 | 26.10 | 41.13 | 39.70 | 41.18 | 38.70 | 43.90 | **45.00** | 44.30 |
| | ARI | 4.29 | 22.81 | 34.10 | 24.50 | - | 41.00 | 43.18 | 40.20 | 45.50 | 45.00 | **46.07** |
| | ACC | 30.00 | 49.00 | 59.30 | 59.50 | 63.00 | 63.10 | 65.90 | 68.10 | **74.00** | 67.00 | 68.43 |
| DBLP | NMI | 15.73 | 20.90 | 26.00 | 26.28 | 32.10 | 33.55 | 38.72 | 38.70 | 43.21 | 40.00 | **44.10** |
| | ARI | 9.03 | 14.00 | 17.20 | 18.00 | 20.00 | 23.80 | 39.00 | 39.20 | 46.00 | 45.28 | **47.55** |
| | ACC | - | - | - | - | - | 29.40 | 30.11 | 30.10 | **31.00** | 28.94 | 28.95 |
| Ogbn-arxiv | NMI | - | - | - | - | - | 40.00 | 43.20 | 44.09 | 44.60 | 44.80 | **45.00** |
| | ARI | - | - | - | - | - | 19.30 | 20.00 | 20.19 | 20.01 | 20.10 | **22.03** |

The experimental results for testing the clustering performance are shown in Table 3. We can see that our method clearly outperforms all baselines in most of the evaluation metrics. However, our advantage is that we are a non-parametric method. That is, we do not need the pre-input $k$ value of parametric methods, and we perform better on graph data than other non-parametric methods. $-$ in the table indicates that the run is out of memory or that the relevant data were not found in the original text.

## 4.3 PARAMETER SETTINGS

In our experiment, the first thing to determine is that $\alpha, \beta$ and $\gamma$ are important balancing parameters. The search range of alpha is $\{10^{-2}, 10^{-1}, 10^0, 10^1, 10^2, 10^3, 10^4\}$. The final selection is shown in Fig. 3. The selection method for $\beta$ and $\gamma$ are same. Depending on the training data, the maximum number of iterations for training is set to 100 to 1000. The pre-training method is to first train the mask network and AE network for 15 rounds, and then jointly train the model for 30 rounds. After pre-training, the learning rate for fine-tuning is set to $10^{-4}$. The dimension of the embedding layer is set to 64. All training is completed on a server with 4 RTX 3080 GPUs.

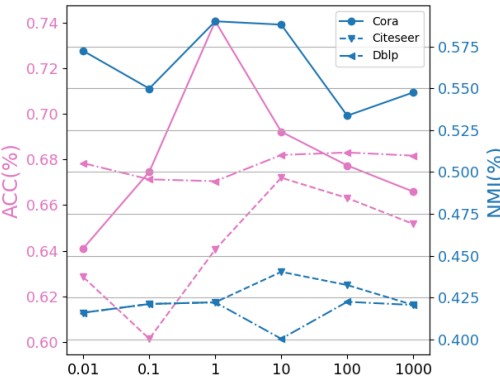

Figure 3: he variation of ACC and NMI on each data set for different values of $\alpha$

## 4.4 ABLATION STUDY

In the previous subsection, we incidentally investigated the impact of fused AE coding on the algorithm as a whole, i.e., the results of GCMA-A. In this subsection, we focus on exploring the impact of the graph autoencoder part and the graph embedding self-optimization part on GCMA. We replace the graph masking part with the autoencoder of the normal GAT layer and adjust the number of factors for the Q-value of the self-optimization part.

As can be seen from Fig. 4(a) and (b), the effect of the GCMA-s model without the self-optimization step is inferior on both datasets. In contrast, in the results Fig. 4(c) and (d) obtained by replacing

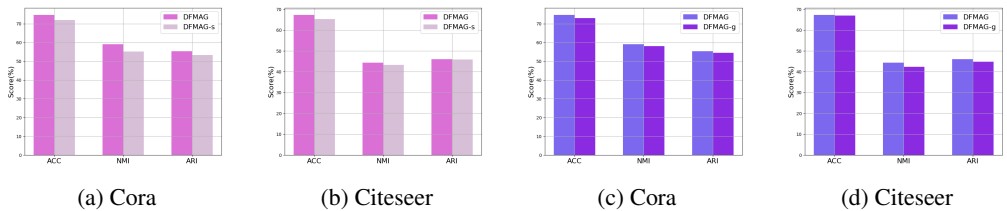

(a) Cora  (b) Citeseer  (c) Cora  (d) Citeseer

Figure 4: (a) and (b) shows the effect of the presence or absence of a self-optimization step on the results. (c) and (d) gives the effect of replacing the mask portion with a normal GAT layer.

the GAT layer, the decrease in ACC values is not significant, but both NMI and ARI values are significantly decreased. This means that the interpretability and generalization performance of the results decreased.

Table 4: $Q_g$ stands for using only $Z_m$ as the $Q$-distribution.The $Q_{ag}$ representation uses $Z_ae$ and $Z_m$. $Q$ stands for the default setting.

Table 5: We conducted experiments using 1 Layer,2 Layer (default settings), and 4 Layer GAT as the basis of the mask and encoder.

Table 6: The effect of different truncation distances $d$. $d_c$ is before improvement and $d_I$ is GCMA's. Experimentation in 10 and 20 rounds.

| Methods | ACC | NMI |
|---|---|---|
| $Q_g$ | 72.64 | 56.60 |
| $Q_{ag}$ | 72.28 | 55.30 |
| $Q$ | 74.74 | 59.16 |

| Methods | ACC | NMI |
|---|---|---|
| 1 layer | 67.00 | 49.80 |
| 2 layers | 74.74 | 59.16 |
| 4 layers | 73.05 | 56.00 |

| Methods | mean | K |
|---|---|---|
| $d_c$ | 7.80 | 7/10 |
| $d_I$ | 7.10 | 10/10 |
| $d_I$ | 7.17 | 19/20 |

Table 7: Performance comparison table for different reconstruction target models. The experiments were performed on two smaller datasets.

Table 8: Generalization ability test results.The selected dataset is similar in size to Cora and Citeseer.

| Methods | Cora | | | Citeseer | | |
|---|---|---|---|---|---|---|
| | ACC | NMI | ARI | ACC | NMI | ARI |
| GCMA-N | 73.88 | 58.68 | 54.90 | 67.00 | 43.91 | 45.82 |
| GCMA-E | 73.15 | 58.60 | 55.00 | 65.06 | 43.86 | 45.57 |
| GCMA-G | 74.70 | 59.06 | 55.10 | 67.05 | 43.90 | 45.93 |
| GCMA-All | 74.75 | 59.10 | 55.30 | 67.33 | 44.32 | 46.04 |
| GCMA | 74.74 | 59.16 | 55.41 | 67.30 | 44.30 | 46.07 |

| Methods | USPS | | HHAR | |
|---|---|---|---|---|
| | mean | Accuracy | mean | Accuracy |
| MtL | 7.0(±2.0) | 5/10 | 7.2(±1.0) | 3/10 |
| DNB | 8.0(±4.0) | 4/10 | 5.0(±0.0) | 0/10 |
| DED | 11.0(±1.1) | 4/10 | 9.0(±2.2) | 6/10 |
| DeepDPM | 9.4(±1.0) | 9/10 | 6.0(±0.9) | 9/10 |
| GCMA | 10.0(±0.0) | 10/10 | 6.0(±0.3) | 9/10 |

Then we also analyzed the reconstructed targets with corresponding experiments. As shown in Table 7. We distinguished the reconstruction targets separately. GCMA-N, GCMA-E, and GCMA-G are used to denote the Node2vec-only target, AE structure-only target, and GAE overall structure-only target. Finally, GCMA-All represents the reconstruction with the goal of combining them together. Based on the data in the table we can conclude as follows: the performance of the three models GCMA-N, GCMA-E, and GCMA-G are comparable, and the performance of GCMA and GCMA-All is better and comparable. However, GCMA has less computational complexity and is more efficient.

Finally, we also explored some experimental default settings in depth. The experiments include the selection of the number of encoder layers, the selection of the sub-distributions of the self-optimizing partial Q-distribution, and the excellent performance test of

Table 9: Generalization ability performance experiment

| Model | GCA | VGAE | GraphMAE | SDCN | GCMA |
|---|---|---|---|---|---|
| Average rank | 3.2 | 4.1 | 2.1 | 4.6 | 1.8 |

the improved and optimized density clustering algorithm. The results are shown in Table 4,5 and 6, respectively. All experiments were obtained by running on the Cora dataset.

To test the generalization ability of our model, we also ran experiments on non-graph datasets (USPS, HHAR). As shown in Table. For the dataset with a missing affinity matrix, we follow

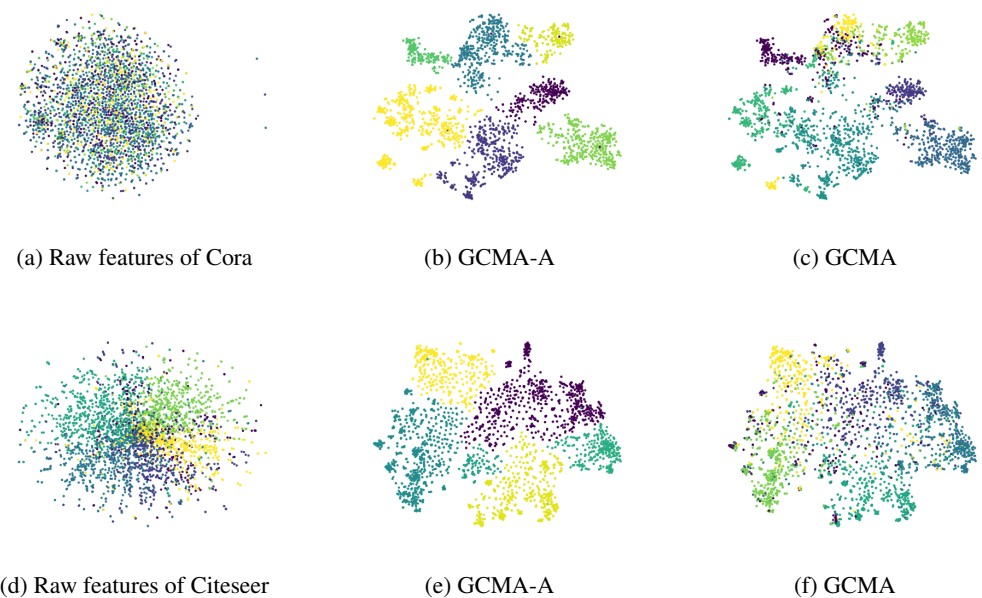

| (a) Raw features of Cora | (b) GCMA-A | (c) GCMA |
|---|---|---|

| (d) Raw features of Citeseer | (e) GCMA-A | (f) GCMA |
|---|---|---|

Figure 5: 2D visualization of 3 datasets: Clustering Process and the Effect of Parameters

(Bo et al., 2020) and construct the matrix using the heat kernel method. As shown in Table 8. The results prove that as a nonparametric class algorithm, our method not only can effectively predict the number of clusters' $k$ value but also shows a performance comparable to the best baseline of the moment. Additionally, we conducted dedicated experiments in Table 9, wherein the model was applied to two distinct tasks—node classification and link prediction. The obtained average rankings are presented in tabular form, clearly demonstrating the robust performance of our model on these novel tasks.

## 4.5 VISUALIZATION OF THE CLUSTERING PROCESS

In Fig 5, in order to visually verify the effectiveness of GCMA,We show a visualization of the clustering process. We can see from the figure that the clustering of GCMA-A is tighter, but the performance in Table 3 shows the opposite result. This is due to the fact that the clustering results obtained are clusters distributed by density, and therefore some sample points that originally strayed from belonging to the original clusters were misclassified. With the addition of the AE fusion coding influenced by the node information, this error is corrected to a limited extent, so that plots such as Fig 5(c) and (f) are obtained.

## 5 CONCLUSION

In this paper, we propose a new graph masking autoencoder framework GCMA. Its core components are our designed fusion image masking autoencoder and improved density-based clustering algorithm. At the same time, a self-supervised method was designed and used to optimize graph embedding. This method encodes and fuses more modal information from both sides, effectively and accurately inducing network training. In addition, the proposed fusion image masking autoencoder can help improve the generalization ability of the model, while also outputting the number of clusters and clustering results end-to-end. We demonstrate the additional value that nonparametric methods bring to deep clustering, namely the sensitivity and importance of hypothesis $k$. Numerous experiments have demonstrated that GCMA performs better than most parametric and nonparametric methods, achieving good SOTA results.

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
