# OpenReview forum: "Graph Clustering with Masked AutoEncoders"
_ICLR.cc/2024/Conference — Submitted to ICLR 2024_

### Official Review · Reviewer_yz1d · 2023-10-14

**Soundness:** 3 good
**Presentation:** 3 good
**Contribution:** 3 good
**Rating:** 5
**Confidence:** 2

**Summary:**

This paper studies the problem of graph clustering and proposes a new framework named Graph Clustering with Masked Autoencoders (GCMA). It involves a graph masking into an auto-encoder framework. Extensive experiments demonstrate the superiority of GCMA over state-of-the-art baselines.

**Strengths:**

- This paper studies a practical problem.
- The main idea of the paper is simple and intuitive.
- The proposed method achieves superior performance on various datasets for different benchmark tasks.

**Weaknesses:**

- The evaluation is not sufficient. The method only involves one large-scale datasets, i.e., Ogbn-arxiv, which is not sufficient to support the claim.
- More result analysis should be included in Sec. 4.4.
- There are some missing prior works about graph clustering in 2022-2023, e.g., [1], which should be included in performance comparison.
- It seems that your self-optimization modules is similar to Deep Embedding Clustering, which should be discussed.

[1] Yi et al., Redundancy-Free Self-Supervised Relational Learning for Graph Clustering

**Questions:**

See weakness.

---

> ### Author Response · Authors · 2023-11-15
>
> **Q1.**
>
> For large-scale datasets, our experimental results on ogbn-arxiv have demonstrated our advantage as a non-parametric method, capable of predicting the number of clusters in large datasets. However, the disadvantage is that, as the data scale increases, hindered by noise interference and dimensionality growth, GCMA, like most methods, experiences a gradual decline in performance. Due to space limitations, we believe it is unnecessary to elaborate excessively on this universally acknowledged aspect in the main text. However, we will provide corresponding experimental results below as supporting evidence. We ran our model on the larger dataset Reddit. While other baseline such as EGAE will have out of memory problem. There is only one way to get the result for the parameter class method S3GC[4].SHOT represents the accuracy of the predicted k-value.
>
> |         | S3GC | GCMA   |
> |---------|--------|--------|
> | Shot    | -   | 2/10  |
> | NMI     | 80.70  | 69.10  |
> | ARI     | 74.50  | 58.98  |
>
> **Q2.**
>
> Additional result analyses have been incorporated into the original text. In Section 4.4, our intention was to demonstrate the superiority of the GCMA method and the impact of integrating AE encoding, namely the GCMA-A model. Therefore, visual results for only two datasets are presented along with brief explanations. From another perspective, Section 4.4 serves as a visual representation of Table 3. It can be observed from the graphs that the separation effect between clusters is worse for GCMA-A, proving the effectiveness of integrating AE encoding. The integrated graph embedding makes full use of node information to correct the clustering results.
>
> **Q3.**
>
> We did not have the methodology of the literature [1] when we started writing this paper, so we added additional experiments. The experimental dataset is cite. However, it is clear that this is a parametric class method that requires the correct number of clusters k to be entered in advance for optimal performance.
>
> |         | R2FGC | GCMA   |
> |---------|--------|--------|
> | Shot    | -   | 10/10  |
> | NMI     | 45.39  | 44.30  |
> | ARI     | 47.07  | 46.07  |
>
> **Q4.**
>
> The self-optimization method we applied indeed stems from deep clustering research, as seen in references [2] and [3]. This method has become nearly a universal approach within the clustering domain for enhancing the quality of autoencoder encodings. It achieves this by compelling better representations through self-supervised training. Subsequent ablation experiments substantiate this point, and for this aspect, we conducted more in-depth experimental research, specifically investigating the impact of the number q on the performance of the self-supervised module (refer to Table 4). The results of these experiments consistently indicate that our application of the optimal number q for the self-supervised optimization module yields the best performance.
>
> [1] Yi et al., Redundancy-Free Self-Supervised Relational Learning for Graph Clustering
>
> [2] Hong Yuan，et al.. Embedding graph auto-encoder for graph clustering
>
> [3]Chun Wang, et al.Attributed graph clustering: A deep attentional embedding approach.
>
> [4]Fnu Devvrit, et al.S3GC: Scalable Self-Supervised Graph Clustering

---

> ### Author Response · Authors · 2023-11-20
> **A reminder**
>
> Dear reviewer, I have submitted my reply a few days ago. Now the deadline for public comment is approaching, but I observe that you have not replied to my comment. I would be happy if you let me know if you still have some questions or replies. If so, I will reply promptly. Looking forward to your reply.

---

> > ### Comment · Reviewer_yz1d · 2023-11-22
> >
> > Thanks for your response. It seems that the paper was not revised.  I will keep my score.

---

### Official Review · Reviewer_CdSX · 2023-10-29

**Soundness:** 2 fair
**Presentation:** 1 poor
**Contribution:** 2 fair
**Rating:** 5
**Confidence:** 3

**Summary:**

The paper proposes a new framework called Graph Clustering with Masked Autoencoders, i.e., GCMA, for unsupervised graph clustering. GCMA employs a fusion autoencoder based on graph masking and an improved density-based clustering algorithm to improve the generalization ability of graph autoencoder clustering algorithms. One advantage of the proposed model is that it can automatically determine the number of clusters k. The authors demonstrate that GCMA can outperform other graph clustering baselines on citation graphs.

**Strengths:**

1. The paper provides a clear introduction to the problem of unsupervised graph clustering and the motivation for the proposed model.

2. A detailed explanation of the fusion autoencoder based on graph masking used in GCMA.

3. GCMA can automatically determine the number of clusters k.

**Weaknesses:**

1. The novelty contribution is incremental. This paper applies graph MAE for graph clustering. However, MAE is well-known and graph MAE has been used for various graph tasks, the idea is not novel.

2. The method description is not entirely clear. As a major difference from the parametric baselines, automatically determine k is a claimed advantage, but how it can be done is not clear from the current writing.

3. Empirical evaluation is not sufficient. Only citation graphs are considered. Does the proposed method also work for other types of graphs?

**Questions:**

1. For Table 3, are the ground truth k used as input for the parametric algorithms?

2. How do the non-parametric methods perform in terms of those metrics in Table 3, e.g., ACC, NMI, and ARI?

3. In section 3.2.2, how exactly was k generated? The paper says "Thus the clustering center can be clearly separated from the rest of the data points to get the best k value". The authors may illustrate more.

---

> ### Author Response · Authors · 2023-11-15
>
> Thank you for your appreciation of our work. In response to your questions, we provide detailed explanations below.
>
> **W1.**
>
> Certainly, as the reviewer pointed out, graph masking methods have become increasingly prevalent. However, there hasn't been a well-performing non-parametric method for graph clustering. In earlier works like MGAE, edge masking was applied to the graph, followed by decoding through cross-correlation. Subsequent methods such as MaskGAE utilized single-edge masking and path masking. More recent approaches like GiGaMAE and GraphMAE introduced feature masking and re-masking to enhance training data for better results. However, current research predominantly focuses on the study of graph masking encoding performance, with downstream tasks mainly oriented towards link prediction and node classification. There is a lack of a dedicated deep learning model applied to clustering tasks. The only existing small-scale downstream experiment on deep clustering with graph masking is found in GiGaMAE, but this model did not incorporate specific improvements for clustering, resulting in relatively poor experimental performance. In light of these research developments, GCMA innovatively combines graph masking technology with clustering. We seamlessly integrate graph masking technology, utilizing single-edge, node, and re-masking, and incorporating AE encoding. We introduce a multi-objective decoder specifically tailored for clustering tasks. In terms of graph encoding technology, our differentiator lies in the modifications made specifically for clustering tasks. We inherit multi-objective reconstruction and the second CFSFDP decoder, achieving better clustering performance while keeping training costs low.
>
> **W2.**
>
> Next, we provide a detailed explanation of the process for predicting the clustering number, k. In the main text, especially in Figure 2, we have depicted the relevant flowchart. The algorithm operates based on two fundamental assumptions. First, data points near the cluster center have lower density, and second, data points are farther away from other centers with higher density. Sample point density is defined as $\rho$, and the minimum distance between a data point and points with higher density is defined as $\delta$. Among all data points, there are points with a product $\gamma_i$ that is larger, representing points with higher density $\rho$. These points are the centers of clusters. After determining several centers, the remaining data points are assigned in order of density, belonging to the point with higher density closest to them. This way, we can automatically obtain the clustering number, k. We integrate this process into our end-to-end GCMA model, optimizing the distance between sample points and cluster centers, further enhancing the algorithm's performance.
>
> **W3.**
>
> Regarding the consideration for dataset selection, our primary concern is the distinction between graph and non-graph data. This allows for better validation of our model's generalization capability. For graph data, most clustering algorithms primarily utilize referenced graph datasets, such as in the literature SDFN, DFCN, S3GC. Rarely do we see other types of graph data in related research.
>
> **Q1.**
>
> For other baselines in Table 3, we provided the correct k value as input. Otherwise, it would be unfair to compare with other parameter-based methods. As mentioned in our Figure 1, the overall clustering performance of parameter-based methods would significantly decline.
>
> **Q2.**
>
> We did not find a non-parametric method that is highly suitable for graph data. Therefore, we had to use non-parametric methods from Table 2 as baselines for performance comparison. However, we acknowledge that this might be unfair to methods not specifically designed for graph data, as they may struggle to showcase their performance on non-specialized data. Here are the additional experimental results we have supplemented. We choose the Cora dataset.
>
> |       | DeepDPM | DED    |
> |-------|---------|--------|
> | ACC   | 11.23   | 4.26   |
> | NMI   | 15.91   | 3.32   |
> | ARI   | 8.10   | 2.13   |
>
> **Q3.**
>
> As mentioned in our response in section "W2," the algorithmic process of CFSFDP allows us to obtain points with larger $\gamma_i$ values, which we consider as cluster centers, thus determining the number of clusters, k. It is essential to highlight the improvements we introduced in our work. First, in our enhancements, we employed a Gaussian kernel to determine the truncation distance. Second, the original CFSFDP, being an independent algorithm, was integrated into the end-to-end GCMA, where we designed a unified loss function for optimization. These improvements position CFSFDP as the second decoding module within GCMA.
>
> Finally, thank you again for taking the time to review my paper and I look forward to hearing from you again.

---

> ### Author Response · Authors · 2023-11-20
> **A reminder**
>
> Dear reviewer, I have submitted my reply a few days ago. Now the deadline for public comment is approaching, but I observe that you have not replied to my comment. I would be happy if you let me know if you still have some questions or replies. If so, I will reply promptly. Looking forward to your reply.

---

> > ### Comment · Reviewer_CdSX · 2023-11-20
> > **Response to authors**
> >
> > Thank the authors for their rebuttal regarding my questions. I partially agree with the response regarding novelty, dataset selection, and non-parametric method comparison. I will keep my score.

---

> > > ### Author Response · Authors · 2023-11-21
> > >
> > > Thank you for your answer, in the previous answer we answered each of your questions in detail but I'm still confused as to why we didn't convince you. If you feel there is something wrong with our approach feel free to continue to ask us questions and discuss it, or if it's unclear we'll go on to give an explanation. But we also need you to give your reasons for questioning so that we can both be better held accountable for what we say. We look forward to your reply.

---

### Official Review · Reviewer_5Nk3 · 2023-10-30

**Soundness:** 3 good
**Presentation:** 2 fair
**Contribution:** 1 poor
**Rating:** 3
**Confidence:** 4

**Summary:**

In this paper, the authors propose Graph Clustering with Masked Autoencoders (GCMA). They claim three different contributions:
1. the first method to determine the number of clusters specifically for graph data.
2. using the mask graph mechanism allows learned representations to be applied to multiple types of downstream tasks
3. Extensive experiments on five datasets demonstrate that our model outperforms existing state-of-the-art baselines.

**Strengths:**

Sorry, I really cannot get significant strengths.

**Weaknesses:**

Originality:
     The whole method is based on GraphMAE (or modified version). The only significant difference is “CFSFDP DECODER“. But it is not new and the authors also point out the citation.
Novelty:
    The novelty is also an issue. Self-supervised learning on graphs + a specific loss on clustering and no experiment on comparison between SSL on graphs and the proposed method.
Performance:
    The comparison only picks up old graph clustering methods and new general clustering methods, not graphs. In my opinion, the performance is fair but not a SOTA one in the graph-based methods. Even I think this method cannot outperform some graph convolution-only methods like
1. NAFS: A Simple yet Tough-to-beat Baseline for Graph Representation Learning

**Questions:**

I have no question. But I am still open to observing how the authors convince all reviewers.

---

> ### Author Response · Authors · 2023-11-15
>
> Thank you for your appreciation of our work. In response to your questions, we provide detailed explanations below.
>
> Firstly, we want to clarify the distinctions between our approach and GraphMAE. GraphMAE primarily masks nodes and their corresponding row vectors, replacing the decoder with a GNN layer (GAT/GIN) instead of an MLP. In contrast, our method masks weights, including nodes and edges, and employs a multi-objective decoding approach to address decoding challenges.
>
> Regarding the previously proposed CFSFDP method, it had three main issues. Firstly, the algorithm's truncation distance, dc, originally required human experience for determination. In our enhancement, we utilize a Gaussian kernel, enabling automatic determination of the truncation distance. Secondly, the original algorithm necessitated the semi-automatic input of the cluster number, k, based on observed built-in parameters. We, however, directly obtain cluster centers by calculating parameters γi and point density. Lastly, the original CFSFDP operated as an independent algorithm, but we integrated it into the end-to-end GCMA, designing a unified optimized loss function. These improvements position CFSFDP as the second decoding module in GCMA. Consequently, our method operates as a non-parametric approach, eliminating the need for pre-inputting the cluster number k. The original method fails to achieve comparable experimental results, and notably, few researchers in the field of graph clustering have employed this method.
>
> Additionally, concerning your remark about not conducting comparative experiments between SSL and the proposed method, this decision stems from the fact that, as a deep clustering model, the SSL module has become a standard component, as seen in AGC and EGAE within our model. Its coercive learning approach has been validated to optimize embeddings to some extent. Therefore, our comparative experiments focused solely on parameter selection within SSL, as detailed in Table 4.
>
> Finally, we would like to explain our motivation for choosing baselines. Our experiments aim to investigate the clustering performance and the accuracy of predicting k for GCMA. The logical design of our experiments is as follows. Two performance metrics are considered: 1) the accuracy of predicting clustering numbers, k, and 2) graph clustering performance. For the former, our experimental baseline is a non-parametric clustering method capable of predicting k. As there are few methods specifically designed for graph clustering, we introduce some deep clustering methods with SOTA performance as benchmarks. For the latter, we need to compare graph clustering performance, thus referencing baseline methods 1, 2, and 3, which are recent SOTA methods. We believe applying graph data to new experimental results obtained through clustering methods would be unfair. The selected graph clustering methods are also among the most effective in recent years, as indicated by various references, including the latest literature [1], which cites these baselines. Experimental results demonstrate that we still achieve considerable performance in graph clustering.
>
> Once again, we appreciate your time in reviewing our paper and look forward to your further feedback.
>
> [1] Yi et al., Redundancy-Free Self-Supervised Relational Learning for Graph Clustering

---

> ### Author Response · Authors · 2023-11-20
> **A reminder**
>
> Dear reviewer, I have submitted my reply a few days ago. Now the deadline for public comment is approaching, but I observe that you have not replied to my comment. I would be happy if you let me know if you still have some questions or replies. If so, I will reply promptly. Looking forward to your reply.

---

### Official Review · Reviewer_sCEC · 2023-10-30

**Soundness:** 2 fair
**Presentation:** 1 poor
**Contribution:** 2 fair
**Rating:** 3
**Confidence:** 4

**Summary:**

This article proposes an unsupervised method for graph clustering based on Graph Masked AutoEncoder. The motivation behind this work is that existing methods struggle to automatically select the number of clusters and lack good generalization ability. To address this issue, the paper introduces a density-based clustering algorithm as the Decoder module. The proposed approach can end-to-end select the number of clusters while improving generalization performance. Extensive experiments demonstrate the effectiveness of the proposed method.

**Strengths:**

- This article addresses a significant drawback of previous graph clustering algorithms, which is the need to pre-determine the number of clusters.

**Weaknesses:**

-**Writing Quality**: The writing quality of this article needs improvement. The language used in the article feels difficult to understand. Additionally, the quality of some of the figures, such as Figure 1, could be enhanced.

-**Method is not clearly presented**:  The introduction of the proposed method in this article is quite unclear. For instance, in Section 3.2.2, the authors abruptly introduce a new concept, CFSFDP, making it challenging for readers unfamiliar with this concept to grasp the purpose of this section. It seems that the authors did not effectively emphasize the main focus of this paper. In my view, since Masked Graph Autoencoders are already established content, the author's primary contribution should lie in the clustering part. However, the authors dedicate a significant amount of content to the introduction of Graph masking and AutoEncoder, which, in my opinion, is unnecessary. While the authors provide Algorithm 1 as a summary of the entire model, this algorithm appears overly concise. Furthermore, in Equation (3), the authors propose using mutual information to calculate the loss. However, I fail to discern the connection between Equation (3) and Equation (4) with mutual information.

-**Rationality for the designs**: Some of the designs proposed in this paper appear to rely heavily on intuition, and certain claims lack robust supporting evidence. For example, in Section 3.1.1, it is not clear why masking both modal information at the same time is considered harmful. In Section 3.1.2, the reasoning behind why a graph autoencoder would overemphasize proximity information needs further explanation. In 3.2.1, the motivation behind adding a mask to Z_m is not well-justified.  I hope the authors can provide their own analysis rather than simply following others' claims.

**Questions:**

Most of my questions have been presented in the Weaknesses.

---

> ### Author Response · Authors · 2023-11-15
> **Weaknesses:Writing Quality**
>
> Thank you very much for your evaluation of our work. We will now provide detailed responses to the raised questions.
>
> We have made several modifications to the original text, improving the grammar and sentences to enhance the overall readability of the article. For instance, in the introduction of the first chapter, we strengthened the logical flow to make the development of graph masking technology more comprehensible. Regarding Figure 1, we redesigned the information within the image. We added a grid to the background and clarified the numerical values of each point, allowing readers to clearly observe the overall trend.

---

> > ### Author Response · Authors · 2023-11-15
> > **Method is not clearly presented:**
> >
> > Our approach is a non-parametric clustering analysis method that primarily employs a graph masking autoencoder. It is crucial to emphasize two key components of our model, GCMA: 1) the masking method and 2) the CFSFDP algorithm. First, introducing the graph masking autoencoder is necessary as it serves as the foundation for the entire GCMA framework, upon which the functionalities of other modules depend. A comprehensive framework introduction is essential for the logical coherence of the entire article. Additionally, compared to graph masking technology, the content related to CFSFDP is more straightforward. Therefore, we did not extensively introduce its original content in terms of length but relied more on the overall flowcharts in the article. We provided more details about the improvements we made to CFSFDP, specifically how to achieve clustering end-to-end and determine the graph clustering number k more accurately. Here, we supplement the content related to the CFSFDP algorithm.
> >
> > The basic assumptions of CFSFDP are twofold: firstly, data points near the cluster centers have lower density, and secondly, data points are farther away from centers with higher density. Sample point density is defined as rho, and the minimum distance between a data point and a point with higher density is defined as $\delta$. Among all data points, there is a type of point with $\delta$ large $\rho$, meaning that their product $\gamma_i$ is larger. Obviously, these points are the centers of the clusters. After determining several centers, the remaining data points belong to the point with higher density nearest to them in sequence. This way, we can automatically obtain the number of clusters, k. In Algorithm 1 of the original text, we have supplemented explanations in this regard.
> >
> > For the second question, we proposed a reconstruction loss algorithm based on Mutual Information (MI) to effectively learn knowledge from multiple collaborative targets. In Formula (3), we explain the computation of the reconstruction loss between two objectives. A straightforward design is to maximize the similarity between the projected representation $S_n$ and the target representation $T_n$. However, this approach learns from each target separately and fails to capture shared information between them, which may include critical common knowledge that the model should emphasize. To overcome this issue, we propose using MI to quantify shared knowledge, specifically the reconstruction process from the masked encoding $S_n$ to the decoded node representation $V_n$, presented in the general form of Formula (3). In Formula (4), we provide detailed explanations of the specific calculation method for Formula (3).

---

> > > ### Comment · Reviewer_sCEC · 2023-11-21
> > > **Why Equation 3 is related to mutual information?**
> > >
> > > Thanks for the response. I am still confused about the relationship between Eq.3, Eq.4 and mutual information. I am familiar with MI estimators such as InfoNCE loss, MINE loss, while Eq.3 and Eq.4 belong to none of them.

---

> > > > ### Author Response · Authors · 2023-11-22
> > > >
> > > > Actually, our loss is an extension based on Mutual Information (MI) and is derived from the InfoNCE loss. Its general form can be expressed as $L_s=L_D(s,t)$.where L  is a lower bound of L(s,t). Specifically, the discriminator D used in L_D is formulated following [1] as $D(s,t)=exp(\frac{1}{\alpha} \frac{f(s) \times t}{||f(s)||\times||t||})$. where $f$ is the projector function, and $\alpha$ denotes the temperature hyper-parameter.
> > > >
> > > > [1] Yonglong Tian, et.al. Contrastive multiview coding.

---

> ### Author Response · Authors · 2023-11-15
> **Rationality for the designs**
>
> First and foremost, it is crucial to clarify that the viewpoints presented in the article are derived from prior relevant research and our experimental results. The perspectives in sections 3.1.1 and 3.1.2 are based on references [1] and [2], respectively. These sections were briefly mentioned in Chapter 1 without elaboration, as they are not the focus of this section, to avoid unnecessary repetition. Therefore, in these subsections, we only provide brief mentions. A more detailed explanation is warranted, as we believe in the viewpoint presented in 3.1.1. This is because, when both node features and edges of the same node are masked, we lack direct information about this node and can only obtain its information from the hidden representations of other nodes. This poses a greater obstacle to the originally intended reconstruction decoding. Without alternative decoding methods, it would have a detrimental impact on the model's performance. To validate this point, we conducted experiments. Specifically, we changed the decoding method to reconstruct the graph structure traditionally. This approach did not consider the harmful effects of simultaneously blocking two modal information types. From the experimental results, it is evident that the clustering performance of this reconstruction method significantly deteriorated. We experimented with the Cora dataset.GCMA-r stands for using the traditional reconstruction graph structure as the decoder, and SHOT stands for the number of correct k-values predicted.
>
> |         | GCMA-r | GCMA   |
> |---------|--------|--------|
> | Shot    | 7/10   | 10/10  |
> | NMI     | 45.00  | 59.16  |
> | ARI     | 36.00  | 55.41  |
>
> Regarding the point mentioned in Section 3.1.2 about GAE overly emphasizing proximity information, this is because previous GAE clustering methods mostly used the link reconstruction graph method to construct loss. In each node's hidden representation, there is information about other neighboring nodes, once again emphasizing the impact of proximity information on overall reconstruction.
>
> As for the point in Section 3.2.1 about re-masking Zm to obtain more compressed representations, this is done because a single masked encoding may not provide comprehensive information for multi-target reconstruction. As the masks are random, using different masked compression representations for the reconstruction of different targets allows our model to learn more diverse modal knowledge content. Additional experiments have been added to demonstrate a significant decrease in experimental performance if we only use a single masked embedding. GCMA-one represents the result under just one mask, and shot represents the number of times the correct k-value was predicted.
>
> |         | GCMA-one | GCMA   |
> |---------|--------|--------|
> | Shot    | 8/10   | 10/10  |
> | NMI     | 57.39  | 59.16  |
> | ARI     | 54.10  | 55.41  |
>
> Finally, thank you again for taking the time to review my paper and I look forward to hearing from you again.
>
> [1] Jintang Li, et al.What’s behind the mask: Understanding masked graph modeling for graph autoencoders.
>
> [2] Yucheng Shi, et al.Gigamae: Generalizable graph masked autoencoder via collaborative latent space reconstruction.

---

> ### Author Response · Authors · 2023-11-20
> **A reminder**
>
> Dear reviewer, I have submitted my reply a few days ago. Now the deadline for public comment is approaching, but I observe that you have not replied to my comment. I would be happy if you let me know if you still have some questions or replies. If so, I will reply promptly. Looking forward to your reply.

---

### Official Review · Reviewer_BL5K · 2023-10-31

**Soundness:** 3 good
**Presentation:** 1 poor
**Contribution:** 2 fair
**Rating:** 3
**Confidence:** 5

**Summary:**

The paper presents GCMA, a framework addressing challenges in graph clustering, specifically generalization and automatic cluster number determination. GCMA utilizes a fusion autoencoder based on graph masking for encoding, combined with an improved density-based clustering algorithm as a secondary decoder. This allows the model to capture more generalized knowledge by decoding mask embeddings. The work emphasizes the importance of determining the correct number of clusters in unsupervised learning and highlights issues with existing methods that overemphasize proximity in graph structures. The authors introduce the graph masking autoencoder to clustering tasks, offering enhanced generalization and interpretability, and through extensive experiments, demonstrate its superiority over existing methods

**Strengths:**

1.  The research problem is significant, as node clustering is a fundamental topic in the graph learning domain.
2. Overall, the paper is well-structured and well-motivated.
3. Extensive baselines are compared in the experimental section.

**Weaknesses:**

1.  The technique novelty is limited.  As they claimed in the introduction, the main contribution of this paper is the usage of graph masking autoencoder for clustering analysis. However, this technique has been well studied in self-supervised learning on graphs/ pretrain models on graphs.  It is not clear why the model can improve the generalization ability.  Another important technique,  density-based clustering algorithm, has also been well-studied in both graph and non-graph domains.

2. Some important claims are not well verified. It claims that the model has better generalization ability. They conduct experiments on noisy/incomplete datasets to verify these claims.  This is confusing.  It is more like robustness instead of generalization.  It is not clear why the model has better interpretability.

3. Some experiments are confusing. For example,  "but both NMI and ARI values are significantly decreased. This means that the interpretability and generalization performance of the results decreased."  How to infer interpretability and generalization from NMI and ARI  are not clear.  From table 3, GCMA is better than GCMA-A in COra. However, the visualization results in Figure 5 show that GCMA-A is better. There is no explanation or description of that.

4. The presentation is messy, especially table 4 to table 8.

**Questions:**

Please check the weaknesses.

---

> ### Author Response · Authors · 2023-11-15
> **Weaknesses:1**
>
> Thank you very much for your evaluation of our work. We will now provide detailed responses to the raised questions.
>
> Our primary contribution lies in the application of graph mask autoencoders for clustering analysis. However, we have not identified any existing non-parametric methods utilizing graph mask autoencoders for clustering analysis. Most clustering models employing self-supervised learning on graphs utilize single-class-based autoencoders, as documented in references [1] and [2]. These references respectively employ GAT and GCN as encoders to construct autoencoder-based deep clustering models. Additionally, certain methods leverage fusion encoders, combining GNN networks and DNN networks, as illustrated in reference [3]. This alignment is also evident in our Table 3, where our chosen baselines are consistent with these methods.
>
> Presently, various masking strategies have evolved in graph mask techniques. The earliest approach, as demonstrated in MGAE, involves initially applying edge masks to the graph followed by decoding through cross-linking. Subsequent methods such as MaskGAE utilize single-edge masks and path masks. More recent models like GiGaMAE and GraphMAE introduce feature masks and heavy masks to enhance training data and achieve superior results. However, current research primarily focuses on investigating the encoding performance of graph mask techniques, with most studies concentrating on downstream tasks such as link prediction and node classification. Notably, there is a dearth of deep learning models specifically applied to clustering tasks. The available data on graph mask methods for deep clustering is limited to a small downstream experiment in GiGaMAE. Unfortunately, this model has not made specific improvements for clustering and exhibits suboptimal experimental performance. Its performance on the Cora dataset is as follows.
>
> |         | GiGaMAE | GCMA   |
> |---------|--------|--------|
> | Shot    | -   | 10/10  |
> | NMI     | 56.39  | 59.16  |
> | ARI     |50.02  | 55.41  |
>
> As a deep graph clustering model, we innovatively integrate graph mask technology with clustering. Our end-to-end integration of graph mask techniques, utilizing single-edge, node, and heavy masks while incorporating AE encoding, involves setting up a multi-target decoder specifically tailored for clustering tasks. In terms of graph encoding techniques, our distinctiveness lies in the modifications made for clustering tasks. We have incorporated multi-target reconstruction and a second CFSFDP decoder, achieving superior clustering performance while keeping training costs economical. For a specialized deep clustering model, our approach marks the first application of graph mask technology, resulting in more compact graph embeddings.
>
> Regarding the density-based clustering algorithm CFSFDP, despite its earlier proposal, it faces three primary challenges. First, the original algorithm's truncation distance traditionally required manual determination based on empirical experience. Our enhancement addresses this by employing a Gaussian kernel, enabling automatic determination of the truncation distance. Second, the original algorithm necessitates the manual input of the cluster number (k) through observation of internal parameters, constituting a semi-automatic determination of k. In contrast, we directly derive cluster centers by computing parameters $\gamma_i$ and point density. Finally, the original CFSFDP is a standalone algorithm, whereas we seamlessly integrate it into the end-to-end GCMA framework and design a unified loss function for optimization. These improvements position CFSFDP as the second decoder module in GCMA. Consequently, our approach, as a non-parametric method, eliminates the need for pre-inputting the cluster number k. The original method fails to yield results as robust as our improved algorithm. It is noteworthy that in the field of graph clustering, few researchers have explored the application of this method.
>
> [1] Chun Wang, et al.Attributed graph clustering: A deep attentional embedding approach.
>
> [2] Hong Yuan，et al.. Embedding graph auto-encoder for graph clustering
>
> [3] Deyu Bo, et al.. Structural deep clustering network.

---

> > ### Author Response · Authors · 2023-11-15
> > **Weaknesses:2,3,4**
> >
> > **Weaknesses:2**
> >
> > Generalization ability reflects the algorithm's adaptability to new samples. We contend that true generalization is demonstrated when a method applied to a specific type of dataset can be used on other datasets of the same type without requiring specific modifications and achieves comparable experimental results. Robustness, as you mentioned, is manifested in the performance on noisy or incomplete data. In literature [4] and [5], relevant studies have shown that graph mask technology enhances generalization ability. In our method, the graph mask encoder is inspired by these findings and adopts a similar structure. The introduction of mask strategies enhances the model's learning capability, theoretically improving generalization ability. The specific reason lies in masking and heavy masking experimental data, which can yield more compressed representation data for the model to learn. This underscores the advantage of mask technology, namely, avoiding excessive reliance on neighborhood structural information and balancing the weights between different pieces of information. Our experiments in Table 8 indicate that the GCMA graph clustering method can also outperform other baselines on non-graph-structured data. Detailed theory and experiments affirm the interpretability of GCMA.
> >
> > **Weaknesses:3**
> >
> > Firstly, addressing the first question, the principles governing the computation of NMI and ARI determine their ability to represent the quality of clustering, thereby indicating the strength of generalization. For commonly used clustering evaluation metrics like NMI and ARI, larger values imply better clustering performance. For instance, NMI is essentially based on the calculation of mutual information, with its components representing the probability of a sample belonging to a certain cluster. The subsequent calculations are derived from maximum likelihood estimation of probabilities. When NMI and ARI decrease, it indicates more evenly distributed allocation probabilities among clusters, making it challenging to achieve an ideal result, i.e., a decrease in clustering algorithm performance. This decrease is due to a change in dataset type, signifying a decline in generalization ability, and the interpretability achieved on the previous dataset cannot be extended to the current dataset.
> >
> > Moving on to the second question, the experimental design in Table 3 aims to validate the clustering performance of GCMA. The rationale behind the ablation experiments in Table 5 is to visualize the impact on GCMA. As seen in Table 3, GCMA performs better, especially in terms of accuracy (ACC). The visualization results in Figure 5 convey a similar message. While the clusters of GCMA-A appear more separated, the absence of the fused AE encoding leads to a decline in its performance. Thus, visually more separated clusters in GCMA-A represent misclustering of some points. The descriptions in these two sections collectively demonstrate the effectiveness of AE fused encoding for GCMA.
> >
> > **Weaknesses:4**
> >
> > Regarding expression, the constraints of small-scale ablation experiments limited our presentation methods. However, for clarity, we aligned tables of different sizes in the original text and merged Tables 4-6 into a single large table to present them neatly.
> >
> > Once again, thank you for taking the time to review my paper. I look forward to your further feedback.
> >
> >
> > [4] Jintang Li, et al.What’s behind the mask: Understanding masked graph modeling for
> > graph autoencoders.
> >
> > [5] Yucheng Shi, et al.Gigamae: Generalizable graph masked autoencoder via collaborative latent space reconstruction.

---

> ### Author Response · Authors · 2023-11-20
> **A reminder**
>
> Dear reviewer, I have submitted my reply a few days ago. Now the deadline for public comment is approaching, but I observe that you have not replied to my comment. I would be happy if you let me know if you still have some questions or replies. If so, I will reply promptly. Looking forward to your reply.

---

> > ### Comment · Reviewer_BL5K · 2023-11-21
> > **Response to authors**
> >
> > Dear Authors,
> >
> > Thanks for the efforts.  I am still not fully convinced by the response.   Specifically,
> >
> > 1.  As the authors have summarized the novelty lies in two aspects.
> > 1) "application" of graph mask autoencoders for clustering analysis.
> > 2) determination of k in deep end-to-end clustering methods.
> > I still believe that the technique's novelty is limited.
> >
> > 2.  The interpretation of model generalization capacity is misleading. Model generalization capacity refers to the ability of a machine learning model to perform well on new, previously unseen data, beyond the specific examples it was trained on. This concept is a crucial and well-established concept in machine learning. Using a model designed for graph-structured data on non-graph structured data and achieving good performance does not necessarily equate to having better model generalization capacity in the traditional sense. This situation is more about the versatility or adaptability of the model to different types of data structures rather than its generalization capacity.
> >
> > 3. "visually more separated clusters in GCMA-A represent misclustering of some points." It is unclear why this conclusion has been drawn.

---

> > > ### Author Response · Authors · 2023-11-21
> > >
> > > **Q1.**
> > >
> > > Thank you for your response. Considering your mention of limited innovativeness, we would like to reiterate that existing methods and relevant literature consistently affirm the absence of a non-parametric graph clustering algorithm utilizing masking techniques for the automatic determination of the clustering number (k). This viewpoint is expounded upon in numerous works, including both comprehensive reviews [1][2] and specialized methodological papers [3][4]. GCMA achieves the goal of automatic clustering number determination through the improved CFSFDP algorithm. Our innovations are distinctive in terms of precision and end-to-end integration.
> > >
> > > **Q2.**
> > >
> > > Regarding the aspect of generalization, we acknowledge the validity of your perspective. However, the elucidation we provided regarding generalization is, in essence, a subconcept of the viewpoint you mentioned—specifically, the ability to perform well on new, previously unseen data. Our focus is predominantly on graph clustering, hence the emphasis on experiments and explanations using both graph and non-graph data. Due to space constraints, we were unable to present this explicitly in the main text. Consequently, we conducted dedicated experiments, wherein the model was applied to two distinct tasks—node classification and link prediction. The obtained average rankings are presented in tabular form, clearly demonstrating the robust performance of our model on these novel tasks.
> > >
> > > | Model     | GCA  | VGAE | GraphMAE | SDCN | GCMA |
> > > |-----------|------|------|----------|------|------|
> > > | Average rank | 3.2  | 4.1  | 2.1      | 4.6  | 1.8  |
> > >
> > > **Q3.**
> > >
> > > In this regard, our intention is to convey that, influenced by the density-based CFSFDP algorithm, the clustering results should ideally exhibit clusters based on density distribution. Consequently, some data points that originally exist outside these clusters might be misclassified. This leads to clusters appearing overly close to each other, and there is a lack of overlap between the samples belonging to the same cluster center and the corresponding cluster. Upon introducing the AE fusion coding, influenced by node information, this misclassification is partially corrected, resulting in patterns such as those depicted in Figures c and f. Notably, points belonging to a specific sample are now closer to other cluster centers, reflecting the corrected outcomes.
> > >
> > >  Looking forward to your reply！
> > >
> > > [1] Malihe Danesh, et al.A survey of clustering large probabilistic graphs: Techniques, evaluations, and applications.
> > >
> > > [2]  Yue Liu, et al.A Survey of Deep Graph Clustering: Taxonomy, Challenge, and Application.
> > >
> > > [3] Hyeonsoo Jo。et al.Robust Graph Clustering via Meta Weighting for Noisy Graphs
> > >
> > > [4] Hong Yuan，et al.. Embedding graph auto-encoder for graph clustering

---

> > > > ### Comment · Reviewer_BL5K · 2023-11-21
> > > > **After author's reponse**
> > > >
> > > > Thank the author for their response.
> > > >
> > > > However, I am not convinced by the author's response.
> > > >
> > > > 1. ``Our innovations are distinctive in terms of precision and end-to-end integration''.  This is quite limited.
> > > > 2.  The usage of generalization is misleading.  I suggest the authors change to another term. The provided results cannot verify the generalization capacity of a model.
> > > > 3. For the third question, to me, from the visualization results, GCMA-A is better.  It seems that the authors don't agree with that and argue that  "visually more separated clusters in GCMA-A represent misclustering of some points.". So my question is how you draw this conclusion from the figure.  I don't find the authors' direct response to my question. It is just an intuitive analysis of the method but not an analysis of the experimental result.
> > > >
> > > > After reviewing the authors' responses, I keep my score.

---

> > > > > ### Author Response · Authors · 2023-11-22
> > > > >
> > > > > 1，Thank you for your response. We are still eager to understand where the limitations lie. As a non-parametric end-to-end method, we have surpassed all the baselines in terms of performance. Additionally, in our ablation studies, we meticulously investigated the impact of each component, affirming the effectiveness of the masking method and the improved CFSFDP algorithm.
> > > > >
> > > > > 2，Regarding what you mentioned about misleading information, I would appreciate it if you could specify the exact areas of concern.
> > > > >
> > > > > 3，In the GCMA-A graph, we can clearly observe a phenomenon where every point near a cluster belongs to that cluster, and there is no overlap between clusters. This is a drawback introduced by density-based algorithms, and it obviously does not conform to the original data distribution. This is what I referred to as misclassified samples.
> > > > >
> > > > > Looking forward to your reply.

---

### Meta-Review · Area_Chair_x1wJ · 2023-12-06

**Metareview:**

This paper presents a framework named GCMA to address challenges in graph clustering, such as improving the generalization ability and determining the cluster numbers. Although the papers studies an important problem and the paper is clearly organized, reviewers raised many concerns regarding novelty, technical contributions, experiments, paper presentation, etc. The authors' responses have addressed some of these concerns. The novelty and technical contribution of this paper is still below the bar of ICLR.

**Justification For Why Not Higher Score:**

The novelty and technical contribution are not significant.

**Justification For Why Not Lower Score:**

N/A

---

### Decision · Program_Chairs · 2024-01-16

Reject